# Motivations of potential anchor businesses to support community development and community health

**Catherine C. Cohen**[1]***, Nabeel Qureshi**[1]**, Raymond Tsai**[2]**, Harry H. Liu**[3]

**1** RAND Health, RAND Corporation, Santa Monica, California, United States of America, **2** Department of Family Medicine, University of California, San Francisco, San Francisco, California, United States of America, **3** RAND Health, RAND Corporation, Boston, Massachusetts, United States of America

* ccohen@rand.org

## Abstract

### Introduction

Some for-profit businesses act like non-profit anchor institutions in contributing to community development, particularly health-related initiatives. Their motives are not well understood. We aimed to 1) identify and describe potential anchor businesses, 2) determine their motivations to contribute to community development, and 3) highlight motivations behind health-related initiatives.

### Materials and methods

We identified a national sample of potential anchor businesses, grouped by those that contributed to 1) both health-related and non-health initiatives, 2) non-health initiatives only, and 3) those without substantial contributions. We conducted an environmental scan, semi-structured qualitative interviews and directed content analysis through rapid review methodology.

### Results

We identified 4,512 potential anchor businesses nationally. Among the 108 of these included in the environmental scan, 48% substantially contributed to community development (12% in health). Interviewees' company philosophies ranged from the idea that economic well-being of the company and community were intertwined, to the idea that commercial success of the company would benefit the community. Motivations for contributions included improving the hiring pool, improved recruitment and retention, and goodwill. Other common sentiments included strategies to focus on core business strengths to address community needs and a desire that companies should not compete in their giving activities. Further, some participants believed health care companies should be investing in health-related initiatives.

### Conclusions

The generosity of potential anchor businesses' local contributions may be determined by company philosophy about its relationship with the community. Stakeholders interested in

**Data Availability Statement:** All relevant data are within the paper and its Supporting information files. We cannot ethically share data from individual qualitative interviews or provide information that would be identifiable by inference given statements

in our informed consent document approved by RAND's IRB. The Donorsearch and Dun & Bradstreet data are proprietary. Location characteristics data are publicly available. We provide a supporting document with data from the environmental scan that is not proprietary or identifiable by inference.

**Funding:** CCC and HHL received a Robert Wood Johnson Foundation grant (grant ID 76909) to fund this work. The funders had no role in study design, data collection and analysis, decision to publish, or preparation of the manuscript. https://www.rwjf. org/en/how-we-work/grants-explorer/funding-opportunities.html.

**Competing interests:** The authors have declared that no competing interests exist.

spurring contributions to local communities might consider messaging to leverage businesses' core strengths and encourage cooperation.

## 1. Introduction

In the last century, large local manufacturers were the engine of the economy, both providing jobs and supporting communities in many US cities and towns. As globalization and manufacturing automation have gradually led to the decrease of manufacturing jobs in the last several decades [1], many communities that once benefited from the presence of large employers became burdened with unemployment, poverty, poor education, and violence [2]. In the absence of a national health system and the ongoing retrenchment of the Federal and State governments [3], however, many local organizations are taking new, innovative approaches to uplift affected communities.

These organizations, often non-profit entities such as universities and large academic medical centers, are considered to be "anchor institutions" because they are rooted in their respective local communities and unlikely to move to another location [4]. These anchor institutions have significant influence on surrounding communities, have non-profit status, and have a social-purpose mission or the potential to acquire one [5]. They help their respective surrounding communities in a variety of ways, including real estate development, directing procurement to local vendors, fostering business creation via direct investment or low-interest loans, offering affordable housing, training local workers, and improving local public education, among others [6]. A prominent example is the West Philadelphia Initiatives launched by the University of Pennsylvania in 1996 to revitalize the University City neighborhood. During 2000–2010, the community's median household income increased, and its poverty rate decreased [7]. Another example is the Greater University Circle Initiative that started in 2005 in Cleveland, Ohio to address persistent poverty and disinvestment in communities. They employed a strategy of "buy local, hire local, live local, and connect" and have made great progress in development [8].

Although less researched, for-profit institutions also increasingly act as anchor institutions and implement initiatives to promote community well-being and culture of health [9]. For example, Cummins, famous for its engines and power generation systems, has spent millions of dollars each year to support the well-being of various communities by supporting technical vocational education among disadvantaged youth, promoting employee community engagement, investing in programs to advance equity for women, and advocating for racial equity [10]. Like non-profit anchor institutions, it is clear that these businesses can have a significant impact on local communities.

While no consensus definition exists for for-profit anchor institutions (hereafter anchor businesses), a conceptual definition was recently proposed: an anchor business is 1) a place-based for-profit entity, 2) unlikely to relocate, 3) has significant local or regional influence, and 4) has contributed to the well-being of surrounding communities [11]. If a business satisfies only the first three criteria, it is considered as a potential anchor business; but a true anchor business must make contributions to community development, that is, beneficial physical or social changes that result in community growth, stability or revitalization.

There is a general consensus in the literature that for-profit and non-profit institutions contributions to community development generate indirect benefits to the companies themselves through improved reputation, employee pride, or local business environment [12]; this is different from individual giving that is often altruistic. However, the financial return from for-

profit business' contributions is often uncertain and difficult to quantify. To differentiate community development contributions from those which serve the purpose of marketing and brand recognition, such as sponsorship, that is part of core business operations, our research focuses on contributions from which a business is unlikely to reap all the direct return. For example, we would exclude establishing new retail clinics by a retail chain, sponsorship of a sports event, or other direct marketing efforts, because these activities are part of core business operations where the intent is to derive a direct return on investments. In contrast, community development efforts we included are often charitable in nature where there is no intended direct benefit to the company's business. Examples could be training the local workforce including those not expected to work for the company, directing procurement to local vendors, building parks, supporting affordable housing, investing in public education, or donating to local food banks.

Factors associated with businesses' contributions to community development have been well researched in the published literature [12]. These drivers include profit and shareholder value maximization [13], executives' moral motives [14, 15], non-owner management [16] or a lack of large stockholders [17], being in consumer-oriented or publicly visible industries [18], and being in industries with environmental or social impacts [19]. However, most literature on this topic is broadly about for-profit businesses, whereas the motivations behind anchor and potential anchor businesses to contribute to community development is less understood.

Given the resources available to for-profit institutions and their great potential to contribute to surrounding communities, it is important to understand which for-profit institutions can be considered as potential anchor businesses, where they are located, and why they make contributions to community well-being. Thus, The Robert Wood Johnson Foundation launched an initiative to understand anchor businesses [20]. As part of that initiative, the objective of this study was to 1) identify potential anchor businesses nationally and describe those that have contributed to community development, particularly around health, 2) determine why some potential anchor businesses have contributed to community development while others have not, and 3) among those with community development initiatives, examine why some contributed to community health and others have not.

## 2. Methods

We operationalized a previously proposed conceptual definition of an anchor business and used the associated criteria to screen potential anchor businesses across the US [11]. We drew a stratified random sample from the potential anchor businesses and conducted an environmental scan to understand the characteristics of the sampled businesses, which were then invited to a qualitative interview to help us understand these businesses' motivations to contribute or not contribute to community development. We detail these multiple methods below. This study was approved by the Human Subjects Protection Committee of the RAND Corporation.

### 2.1. Identifying potential anchor businesses

To identify companies that could be considered potential anchor businesses, we first identified 30 exemplars of for-profit businesses thought to be anchor businesses in published literature or identified in expert consultation using a conservative principle: if this is not an anchor business, anchor businesses do not exist. Next, using 2020 Dun & Bradstreet data, determined up to 5 for-profit companies within the industry and headquarters' Metropolitan Statistical Area (MSA). The Dun & Bradstreet database collects and organizes data on key metrics of

businesses in the US, including their revenues, number of employees, and organizational structure (i.e., whether a company is a subsidiary, its for-profit status, etc.). Then, we compared these anchor businesses and their matched pairs to refine and operationalize the definition considering both relative and absolute measures of employment, revenue, donations and industry mobility (see Table 1 for data sources).

We determined those anchor businesses' respective economic footprint in their headquarters' community differed from matched pairs in the proportions of employment and revenue for which the company was responsible within the company headquarters' MSA, i.e., the employment ratio—the number of company employees at the headquarters to total working population of the headquarter's MSA, and the revenue ratio—that ratio of total company revenue to total revenue in MSA. As the goal was to include a vast majority of potential anchor businesses, we considered them to be a potential anchor business if a business met a threshold that was either the 25th percentile of the example anchor businesses' employment ratio or the revenue ratio (0.016% and 0.034%, respectively). Although a company's mobility is part of a previously proposed anchor business definition, it was not possible to include this criteria in our study due to a lack of an accepted measure, and lack of data on whether a business is a placed-based entity.

Further, to quantify relevant community contributions from these exemplars and matched companies, we used the DonorSearch data base (www.donorsearch.net) to determine the average annual company contributions during 2015–2019 for these companies. The DonorSearch database collects information about which organizations donate to charitable causes or organizations and the amount they donate as a part of their philanthropic efforts. Data is categorized by types of donations (i.e., political donations, annual donations, in kind donations, etc.) with information on the source of the data (typically, the organization's annual report). We focused on contributions to the community surrounding their US headquarters, excluding political donations and those made from company foundations, as contributions from these foundations may represent funds obtained from other entities in addition to the company. We summed the contributions meeting our criteria for each company and created annual averages. Based on the distribution of non-political charitable contributions, the median contribution of $50,000 appeared to be a reasonable threshold for what may constitute substantial contributions. We used this threshold in sampling and recruiting (described below).

To screen for potential anchor businesses nationally, we applied the employment ratio and revenue ratio to the 2020 Dun & Bradstreet database. We examined the distribution and characteristics of companies meeting these criteria across the US.

## 2.2. Environmental scan methods

To identify the businesses for our environmental scan, we took a random subsample of potential anchor businesses, stratified by urbanization level (i.e., the population size of headquarter

**Table 1. Study data sources*.**

| Concept | Source | Year(s) |
|---|---|---|
| Community Investment | DonorSearch, Environmental Scan | 2015–2019, 2020 |
| Company Policies | Environmental Scan, Qualitative Interviews | 2020, 2021 |
| Employment | Dun & Bradstreet, U.S. Census Bureau | 2020, 2017 |
| Revenue | Dun & Bradstreet, U.S. Census Bureau | 2020, 2012 |
| Industry Mobility | Duranton (2007) [21] | |
| Giving Motivations | Qualitative Interviews | 2021 |

*NOTE: All sources represent the most recent data available at the time the study was conducted.

MSAs), and level of spatial immobility based on industry [21], given that some studies have linked company contributions with industry type. We developed three urbanization levels (greater than 1 million people, between 250 thousand and 1 million, and less than 250 thousand) and three levels of immobility based on terciles of the underlying sample. We randomly sampled 12 companies from each of the 9 strata resulting in a total of 108 companies. This sample was thought large enough to ensure a pool of company candidates for each stratum for a deep-dive analysis. We replaced companies that had gone out of business or were mislabeled with other randomly selected companies in the same stratum that have similar employment ratios, revenue ratios, and are based in a location with a similar urbanization level. We conducted a detailed, systematic environmental scan to confirm businesses' economic ties and community development contributions, among other characteristics. To confirm economic ties to the community, we collected information regarding whether the company hired local, sourced their supplies locally, supported local business development or participated in their local chamber of commerce. To determine contributions, we gathered information from the DonorSearch database to identify all non-political donations from the companies to the community surrounding their US headquarters during 2015–2019. We also sought and recorded any company contributions not captured by the DonorSearch database. Further, the environmental scan collected information related to the company's mission, values and a sustainability plan as anchor institutions have social purpose missions and anchor businesses may use similar language. We further collected information that we hypothesized might affect companies' willingness to contribute and was publicly available, such as whether the company is family-owned or a co-op.

We conducted internet searches and reviewed each company's US website, online publications, and annual reports for the above information for our systematic environmental scan, including public companies' filings to the Security Exchange Commission, where applicable. To ensure the data were collected systematically, we developed a standardized data collection tool and a corresponding protocol (See S1 File), which promoted consistent data abstraction from diverse websites and other sources as well as consistent interpretation of the tool items by data abstractors (i.e., operational definitions). Three reviewers on the research team piloted the tool and protocol with the first 10 companies of the sample to assess agreement between reviewers. The tool and protocol were refined by discussion to consensus, further ensuring complete and consistent data collection throughout this process.

## 2.3. Qualitative interviews

Companies that made substantial annual contributions ($\geq$\$50,000) represented 45.4% of those in the scan sample; this enabled us to classify companies to different groups for interview recruitment. Thus, we classified potential anchor businesses from the environmental scan into the three categories: those that have made substantial contributions to support community health and/or other community development programs (group 1: contributors to both health and non-health initiatives), those that made substantial contributions to community development, but not directly related to health (group 2, contributors to non-health initiatives only), and those that have made investments less than \$50,000 annually to the community (group 3, low-contributors).

To ensure we would interview comparable businesses across the 3 categories outlined above, we started our selection of companies to recruit by randomly selecting 4 businesses from a stratified sample from group 1. Once the interview was scheduled, we then attempted to recruit businesses in groups 2 and 3 that matched the same industry sector and urbanization

level to those identified in group 1. We targeted a total of 12 companies with 4 companies in each of the 3 groups.

In January-July 2021, we conducted recruiting outreach via email templates and phone scripts, using contact information provided in the Dun & Bradstreet database. If the contact information from the Dun & Bradstreet database was missing or not valid, we searched company websites including public reports related to corporate social responsibility (CSR) and LinkedIn for contact information. For all companies, we called and requested to speak with at least one employee who was knowledgeable about their business mission, local relationships, and contributions to community development and therefore holds a title such as director of philanthropy, vice president of corporate and community philanthropy, or director of public relations. Individuals from philanthropy departments were included because they oversee the giving activities of the company, even though philanthropy dollars were not included in assessments of anchor businesses due to the ability for philanthropy dollars to include company funds as well as funds from other individuals or organizations.

We conducted semi-structured interviews over Microsoft Teams video conference following a discussion guide intended to steer a 30–60 min conversation with one or more participants per company (See S2 File). We conducted pilot interviews with three companies outside of our sample that had substantial philanthropic contributions that both included and excluded health-related initiatives. The goal of these pilot interviews was to refine our interview protocol. Participants were offered $150 per 30-minute interval for participation. With participant permission, we audio-recorded these discussions and transcribed the interview recordings. We de-identified transcripts, organized notes, and extracted relevant quotes and emerging themes using our discussion guide as the initial framework, following a rapid review analysis [22]. We made note of new philosophy and contribution information not captured in the environmental scan, including contributions from foundations.

## 3. Results

Applying our screening criteria to the Dun & Bradstreet database identified 4,512 potential anchor businesses nationally. There were potential anchor businesses in all 50 states, reflecting 1,160 counties (see S1 Fig). Roughly 69% of US counties have less than 5 potential anchors businesses, and ~96% have less than 11. Among the counties with at least one potential anchor business, there were between one and 47 anchor businesses per county, for an average of 3.9 (SD = 3.4) and a median of three per county. The highest concentration of potential anchor businesses were in the West and Midwest. About a quarter of companies were in manufacturing, while 22% were in the service industry, 18% in retail trade, and 12% in finance, insurance, and real estate according to Standard Industry Classification codes [23]. The sampling process and results for the environmental scan and interview recruiting is presented in Fig 1.

### 3.1. Environmental scan sample description

As shown in Table 2, in our sample of 108 potential anchor businesses, the majority of companies were part of their local chamber of commerce (81%). Very few companies had explicit policies to hire locally (7%), source supplies locally (6%), or support local business development (4%). Few companies were family owned (14%), were co-operatives (6%), or owned by a foreign entity (6%). Overall, 4% had a website devoted to corporate social responsibility (CSR) activities and 13% of companies had specific policies around environmental sustainability as a part of their Environment, Social, and Corporate Governance (ESG) goals.

A total of 48% of the environmental scan subsample had contributed to community development, including in health (group 1), 12% had contributed to community development, but

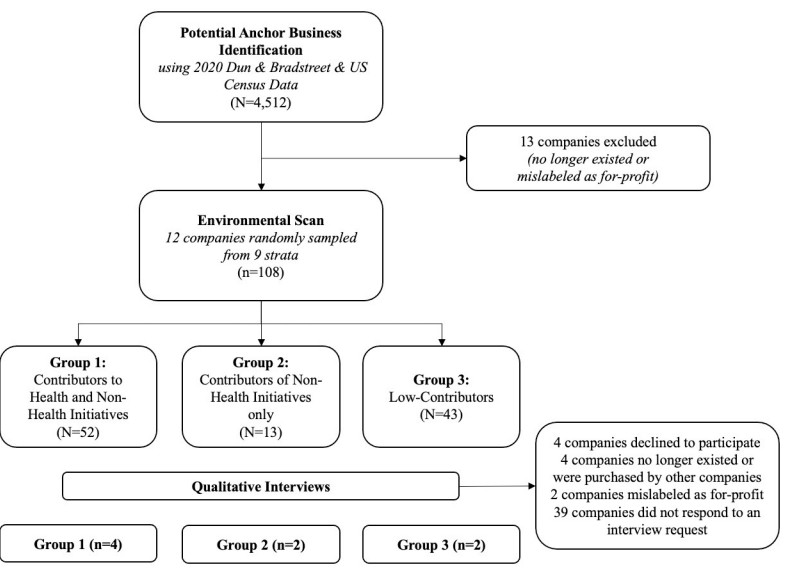

**Fig 1. Sampling of companies for environmental scan and interviews.**

**Table 2. Environmental scan sample description.**

| Characteristic | | Group 1: Contributors to Health and Non-Health Initiatives (N = 52) | Group 2: Contributors of Non-Health Initiatives only (N = 13) | Group 3: Low-Contributors (N = 43) | All Companies (N = 108) |
|---|---|---|---|---|---|
| Company | % Family Owned | 12% | 23% | 14% | 14% |
| | % Co-op | 2% | 0% | 12% | 6% |
| | % Foreign Ownership | 4% | 0% | 9% | 6% |
| Economic Influence | Avg. % of MSA Working Population Employed at Headquarter | 5% | 2% | 2% | 3% |
| | Avg. Total Employees | 53,118 | 32,482 | 16,392 | 36,012 |
| | Avg. Revenue Ratio | 0.14 | 0.05 | 0.92 | 0.44 |
| | Avg. Total Revenue (in billions) | $21.52 | $4.37 | $2.95 | $12.07 |
| | Chamber of Commerce member | 88% | 69% | 77% | 81% |
| Immobility | Avg. Industry Mobility Index | 0.67 | 0.70 | 0.67 | 0.68 |
| Giving | Median Annual Donations | $414,385 | $64,494 | $0 | $19,463 |
| | Max. Annual Donations | $241.72 M | $1.39 M | $1,762 | $241.72 M |
| | Avg. Annual Donations | $13.32 M | $293,380 | $117 | $6.45 M |
| Mission & Values; % with company mission and materials including: | Environmental Sustainability Plan | 15% | 8% | 12% | 13% |
| | Hiring Locally | 6% | 8% | 9% | 7% |
| | Sourcing Supplies Locally | 0% | 0% | 14% | 6% |
| | Supporting Local Business Development | 6% | 8% | 0% | 4% |
| | Health Focus | 29% | 15% | 12% | 20% |
| | Justice & Equity Focus | 2% | 0% | 2% | 2% |
| | CSR Webpages or Reports | 6% | 0% | 2% | 4% |
| | Mission Statements | 98% | 85% | 70% | 85% |

Note: Avg. = Average; M = Million; Max = Maximum; CSR = Corporate Social Responsibility.

not in health (group 2) and 40% did not appear to have substantial contributions to local community development (group 3). Of those who made health-related contributions, 29% had a specific focus on health in their mission statements (such as improving access to care or insurance), as opposed to 15% of those who only made non-health contributions and 12% of making minimal contributions (Table 2). In an absolute sense, companies in group 1 had more employees, higher revenues, and greater average annual donations than those in group 2 or 3 (Table 2). No additional company donations were revealed by the environmental scan that did not appear in the DonorSearch database. S3 File shows deidentified and non-proprietary environmental scan data.

### 3.2. Interview respondent description

As described in Table 3, we interviewed representatives from eight companies. Nearly all interview respondents were company executives. These individuals occupied positions in CSR, Philanthropy, Community Affairs or Relations, and Public Affairs. One respondent was also in the Department of Investor Relations. Hereafter, we refer to interview respondents by group and numerical company identifier, for example, "non-health contributor 1" indicates company number 1 within the group making substantial contributions, but not to health-related community development.

Of the eight companies represented, two were primarily in the banking industry, two were in the health industry, two were in the insurance industry, and two were in commercial sales or retail services. Two companies were family-owned. Three companies were associated with a philanthropic foundation, including one low-contributor. We determined the foundation of

**Table 3. Sample characteristics of interview respondents and non-respondents.**

| Characteristic | | Respondents (N = 8) | Non-Respondents (N = 49) | All Companies Selected for Recruitment (N = 57) |
|---|---|---|---|---|
| Company | % Family Owned | 25% | 16% | 14% |
| | % Co-op | 0% | 6% | 5% |
| | % Foreign Ownership | 0% | 6% | 5% |
| Economic Influence | Avg. % of MSA Working Population Employed at Headquarter | 5% | 2% | 3% |
| | Avg. Total Employees | 48,679 | 33,576 | 35,696 |
| | Avg. Revenue Ratio | 0.09 | 0.87 | 0.76 |
| | Avg. Total Revenue (in billions) | $7.32 | $13.73 | $12.84 |
| | Chamber of Commerce Member | 75% | 73% | 74% |
| Immobility | Avg. Industry Mobility Index | 0.71 | 0.64 | 0.65 |
| Giving | Median Annual donations | $40,089 | $10,000 | $17,109 |
| | Max. Annual Donations | $4.08 M | $241.72 B | $241.72 B |
| | Avg. Annual Donations | $657,181 | $7.10 M | $6.20 M |
| Mission & Values; % with company mission and materials including: | Environmental Sustainability Plan | 25% | 12% | 14% |
| | Hiring Locally | 0% | 6% | 5% |
| | Sourcing Supplies Locally | 13% | 8% | 9% |
| | Supporting Local Business Development | 13% | 2% | 4% |
| | Health Focus | 38% | 20% | 23% |
| | Justice & Equity Focus | 13% | 2% | 4% |
| | CSR Webpages or Reports | 12% | 2% | 4% |
| | Mission Statements | 100% | 76% | 79% |

Note: Avg. = Average; M = Million; Max = Maximum; CSR = Corporate Social Responsibility.

that low-contributor company had made substantial donations, but not in health-related community development. In light of this information, we recategorized this company as a non-health contributor for our qualitative analysis. Six of the eight companies interviewed donated substantially to community development, four of which contributed substantially to health-related initiatives.

**3.2.1. Community development contributions.** In general, companies we considered to be contributors donated $657 thousand per year on average to community development directly from their businesses. Among companies contributing to health-related initiatives, average contributions were $1.05 million on average. Half of our sample sold their goods or services through a direct-to-consumer model (all health contributors). Of the four other companies, one sold insurance through brokers, one was a wholesaler to insurance companies and healthcare facilities, one was a wholesaler to restaurants and other food service providers, and one had a franchising model.

**3.2.2. Mobility.** In the interviews, company representatives perceived that their company was rooted in their respective headquarters' community. When asked how difficult it would be financially or otherwise for the company to relocate to a new community, some said they did not know, it depends or it was "evolving" due to COVID pandemic and remote work (n = 4). Two declared that the company could never fully leave, even if the headquarters moved as some operations or customers would remain. The representatives of low-contributor companies said moving would not be "culturally . . . realistic" to "virtually impossible". Thus, while only half the company representatives thought leaving the headquarters community was extremely unlikely, none responded that a move would not be difficult.

**3.2.3. Economic ties.** All the companies thought they were really important to their respective headquarters' community. When asked what would happen to the surrounding community economically if the company moved to a new location or closed, some answered "it would rock [the community] to its core" (low-contributor #1) and "there will be a pretty significant job loss . . . the ripple effect of losing that many jobs . . . would be very devastating" (low-contributor #2). Others responded that there would be some local economic disruption.

## 3.3. Business' motivations for contributing to community development

We determined that all the companies represented in our study did some philanthropic work within their local communities, including those deemed to be low-contributors. Thus, we were able to ask all interviewees about their company's social responsibility or community development initiatives and understand their motivations for such contributions, as well as why the company prioritized specific issues, such as community health.

**3.3.1. Company community development philosophies.** When asked to describe their company's beliefs about CSR and a business' role in community development, if any, answers ranged from being largely philanthropic, having both philanthropic and commercial motives, to being primarily commercially-driven. The two low-contributor companies were those primarily commercially driven (see relevant quotes in Table 4).

Those with more philanthropic philosophies used the terms "responsibility", mentioned "promises", and that the company feels "deeply tied to [philanthropy]" in the community. In particular, one company representative said they purposely changed their written philosophy language away from "social responsibility" because they do not want to communicate a sense of obligation to have a social impact, but rather that their giving is driven by what the community needs (non-health contributor #2).

For the companies that have both philanthropic and commercial motives in contributing to community development, they often have a clear strategy to integrate community

**Table 4. Themes and sub-themes regarding motivations for community development contributions among potential anchor businesses and exemplar quotes.**

| Theme | Sub-theme | Exemplar quote |
|---|---|---|
| Company Community Development Philosophies | Philanthropically-driven | "We consider ourselves an integral part of the community. And as such, we need to be investing in the community. . .I would say we very much view it as part of who we are as part of our culture. Part of our DNA. We make our decisions based off of that." (health contributor #3) |
| | Synergistic Community and Company Success | "We do believe in that shared value of things that support the customer, the communities, the associates and our shareholders. No money, no mission. . . . we need to have revenue to give away to the community. And it's an ecosystem that we need to build on." (health contributor #1) |
| | Commercially-driven | "We're big believers . . . in having to create profits so that we can survive as a company." (low-contributor #2) |
| Motivation for Contributions in General | Benefit The Company Through Goodwill Or Human Capital | "[Contributions make employees] proud to say that they're part of an organization that has a positive reputation within their community and that just in turn helps us attract and retain the best in class employees." (non-health contributor #1). |
| Motivation For Health-Related Investments | Contribution History | "We have a lot of relationships [with non-profit partners] that we built over time . . . I like to say is we're really bad at dating, but . . . we're pretty good at getting married, . . . we're married forever." (low-contributor #2) |
| | Leveraging Core Business Strengths | "It's using the strength of the core business in community development. You're focusing in on the things you know you do well." (health contributor #4) "Illnesses and medical, stability or even medical advocacy or health advocacy. It's not our strength" (health contributor #1). |
| | Impact of Investments | [General] "Our focus [of community development] is trying to find that sweet spot of where our business can make a difference." (health contributor #3). |
| | | [Health] "This high failure rate in some of the experimentation is critical. . . .Well, the medicine didn't work or whatever the case may have been. And people's tolerance may not be as high on those things. You know, versus we built housing. We built it or we didn't." (health contributor #1) |
| | Ecology of Giving | "We don't tend to give to the health industry as much. That's a really, really vast array of giving and I mean you've got to have your pillars in certain areas and we've got other businesses in the area where their focus is health." (non-health contributor #1) |
| | Community's Perceived Needs | "So you look at [community development] from a market plan approach and understand like what seems to be driving the need." (health contributor #4). |
| | Did Not Know Rationale | "I can't really speak to how [company leaders] go about selecting [community development initiatives]" (non-health contributor #1) |
| Barriers To Community Development | Communicating Impact | "It's hard to tell people like you made a difference. . . .The return of my $1 million." (health contributor #1) |
| | Constrained Resources | "It's not an endless pool of money, and so. . . it's deciding those priorities" (health contributor #2) |
| | Partner Selection | "The two hardest things that we work on issue identification and partner selection." (health contributor #3). |
| | Multiple, Competing Priorities | "What is that right marriage between an organization that is within our [core business's] scope, that wants to kind of be geographically dispersed, perhaps that's tackling the issue in a way that's not just putting a band aid on something, but that's also trying to you know, solve some of these issues and how does [our company] fit into that? Because like I said, I know we're not going to write the biggest check, . . . how do you even find that kind of opportunity?" (low-contributor #2) |

development initiatives with their commercial interests. For instance, one company (health contributor #1) emphasized the shared value between customers, communities, the company itself, and the whole ecosystem. Another company with a similar giving philosophy (health contributor #3) deliberately chooses community development initiatives that are in line with their company strength, business gaps, and market opportunities.

In contrast, representatives of two companies with more commercial missions responded to the same question with statements such as needing to "keep the costs as low as possible [to

consumers]" and emphasized direct benefits to core business operations and a company's commercial success.

**3.3.2. Motivation for contributions in general.** Some companies said a key consideration was "trying to find ways to support nonprofits and groups, [. . .] to have some sort of activity that we think will come back and benefit the company" (low-contributor #1). The way that these companies hoped to ultimately benefit from contributions to community development was through either human capital or goodwill among investors, the community and other stakeholders.

Human capital benefits were described either through future workforce development or employee morale. For example, three company executives talked about improving the local hiring pool. Specifically, they invested in scholarships or education programs that would help future employees build "on [a] strong foundation" when they come to work at the company (health contributor #2). Other companies invested to promote employee satisfaction, morale and the company culture and thus their community development efforts were "tied to talent and recruitment" (non-health contributor #3). Two participants (representatives of one health and one non-health contributor) went further saying "our responsibility as a company is to our team members" (low-contributor #2) and discussed how specific community development initiatives were perceived to be popular among employees, thus supporting their company culture and morale.

In terms of generating goodwill among investors, the community and other stakeholders, company representatives explained that by engaging in certain activities, it allows the company's name to be "recognized in the local community [and builds a] broader public understanding about what we do." Eventually, the company can use this general goodwill to advocate for public policies that benefit both the company and the community (low-contributor #1).

## 3.4. Priority setting for contributions, especially health-related initiatives investment

Companies weighed a balance of considerations to prioritize contributions for specific community development activities. When asked about motivations and considerations for health-related community development investments, some participants did not know exactly why health was or was not a priority, while others offered detailed thoughts on this topic. For example, one could not definitively say why health-related community development was not a priority, but suspected that it was not seriously considered given conscious decisions towards another priority area (for instance, environmental sustainability) that ultimately benefits the company (low-contributor #1). These individuals also expressed that their respective companies had a policy to focus on priority topics for giving that did not include health.

Among the other participants, responses differed regarding the motivating factors, considerations, and barriers to investing in health-related community development. These responses included contribution history, leveraging core business strengths, investment impact, a giving ecology, and the community's perceived needs.

**3.4.1 Contribution history.** History of giving priorities was a barrier to health contributions. A representative of a low-contributor said given their focus and success with other priority areas would make initiating health-related community development would be "working a little against the current" and added "We have been able to make a meaningful impact. It would be unnatural to introduce a new subject matter" (low-contributor #2).

Momentum was also a driver of giving priorities beyond health-related initiatives. Two representatives mentioned their companies' "list of giving, especially from a sponsorship side doesn't really change a lot. . . year over year, [it is] basically the same" (non-health contributor #1).

### 3.4.2. Leveraging core business strengths.

Being able to leverage core business strengths was a driver of both health- and non-health-related community development investments. One company had recently expanded operations to a health-related field. Their newly developed expertise in that area came with an understanding of the gaps in access to care, which motivated their new community development initiatives (health contributor #2). Another non-health company mentioned that they use skills of the core business to help with a health-related initiative in partnership with a charitable organization. Expertise used were in marketing, technology, commercial small business strategy and market analysis to advise local health and non-health-related non-profits to improve their operations (health contributor #1). However, this company said health giving is not a priority because it is not aligned with their core business, despite that by our measure, they had substantially contributed to health.

The ability to "[leverage] what our expertise is as a company" was also the most widely-mentioned motivator for community development investments in other specific priority areas and issues (non-health contributor #2, low-contributor #2).

### 3.4.3. Investment impact.

Representatives mentioned that companies prioritized community development issues, both in health and non-health areas, based on the expected impact of those contributions. Specifically, two companies mentioned impact of investments was a barrier to health investment, especially the difficulty of measuring the impact of health-related contributions (health contributor #1, health contributor #4) due to a long time horizon of the investment and the uncertain return on investments (health contributor #1) in health-related community development.

That said, these companies had both substantially contributed to health-related initiatives and thus were not fully deterred by these barriers, but they did not consider themselves to be focused on traditional health-related community development initiatives.

The desire to maximize impact was an important consideration for broader community development initiatives as well, not just those in health. Companies want to be a "catalyst for change" (health contributor #2) and representatives expressed companies' desire to "[tackle an] issue in a way that's not just putting a band aid on something" (low-contributor #2).

### 3.4.4. Ecology of giving.

When potential anchor business representatives discussed how health-focused and non-health focused community development differs, the most common concept described was a desire to fit within an ecosystem of local charitable giving (health contributor #1, non-health contributor #1 and health contributor #4). Within a geographic area, companies wanted to "coordinate resources in communities to make sure that all the investment is done effectively" (health contributor #4). That is, they wanted to avoid overlapping or competing with other companies' investments in the same community and further "to successfully leverage each other" (health contributor #4). That said, these potential anchor businesses, including one health contributor had a perception that health initiatives should be the responsibility of other companies for which these initiatives would be more aligned with a business "strength".

However, one company in the health industry, who did not make substantial contributions to health-related activities by our measures, stated that they do not make health-related contributions through their foundation either because they are "super cautious" to avoid community development activities "that could be perceived as a conflict [of interest]" with their core business (non-health contributor #2).

### 3.4.5. Community's perceived needs.

When asked why their company invests in specific initiatives and not others, representatives of companies discussed how they had a desire to be "responsive to what the community needed" (health contributor #3) by aligning investments with the most pressing needs. In this regard, company representatives said they "look at it from a market plan approach and understand like what seems to be driving the need" (health

contributor #4). Many times, this would rely on doing "our research [. . .] to know what the issues are" (health contributor #2), and then thinking about what strengths the business could use to address those problems.

## 3.5. Barriers to community development

When asked what the barriers were to investment in community development, company representatives mentioned issues that they perceive to be limiting the impact they can make with these initiatives. These include constrained resources, partner selection and having multiple competing priorities.

Participants most frequently mentioned constrained resources as a barrier to giving. Quantifying impact was again mentioned as a specific barrier (health contributor #1). Another mentioned the tension between investing enough to impact a particular area and not neglecting other areas where the company has a physical presence. That is, this work has to be done at a very local level, but the company also needs to avoid the "tension of being overly invested in [where the headquarters is located]" (low-contributor #2).

Other issues included researching potential partners, coordinating resources with them (health contributor #4), and connecting to collaborators with similar motivations (health contributor #3). One company specifically mentioned that the pandemic has made this selection process harder due to a lack of in person interactions (health contributor #2). Finally, a company representative mentioned that bad experiences with new partners was a barrier to new investments in general and their preference to "stick to relationships that they know works." (low-contributor #2)

This same representative emphasized that it was a significant challenge to find a community development investment issue or opportunity that simultaneously met all the priorities around meeting community needs, leveraging business core strengths, with high potential for impact in light of dispersed operations and limited resources (low-contributor #2).

## 4. Discussion

This study is the first to determine the number and distribution of for-profit companies that, relative to their local communities, appear to have the economic scale to be anchor businesses. These large businesses may hold the key to expanding contributions to community development, in particular in health-related initiatives. About a third of counties in the US had at least one anchor business, with the highest concentration of potential anchor businesses in the West and Midwest and roughly equal numbers of companies in manufacturing and the service industries. That said, these potential anchor businesses appear to be widely distributed across the US, but not concentrated within counties, which may have implications for companies' desire to be part of an "ecology of giving", as described by our interviewees.

Overall, our finding that these potential anchor businesses were motivated to contribute to community development in part to support their businesses was not surprising. When company representatives described their goals of such contributions, they often implicitly, sometimes explicitly, mentioned how these community development initiatives could benefit their businesses, such as improving their reputation and attracting talent. These findings regarding potential anchor businesses were similar to those from prior literature about general for-profit companies' motivations for contributing to community development [12].

This study adds that a company's philosophy about its relationship with the community appears to affect the extent to which they contributed to community development. Our results show that some businesses consider themselves a part of an ecosystem and perceive obligations to support the surrounding community, especially in the areas they have a reasonably large

presence. Given the need for for-profit companies to be profitable, it is not surprising that some companies viewed community development activities as being dependent on revenue and that there is a symbiotic relationship between the community and their business's success. But, there appears to be a continuum of company philosophies among these potential anchor businesses ranging from being purely philanthropically motivated to simply commercially driven. The two low-contributor companies with mostly commercial motives did not contribute substantially to community development, whereas six companies with a philanthropic or both a philanthropic and commercial motive did. Therefore, it seems companies driven primarily by commercial motives are less likely to contribute substantially to community development than those driven both commercially and philanthropically or primarily by philanthropic motives.

Of note, it happens that these two commercially driven companies are also operated by founding families; this is consistent with prior research on ownership structure's impact on philanthropic giving [16]. Companies managed by owners or with large stockholders (e.g., institutional investors) are less likely to be big corporate contributors to community development.

With regard to the motivations to contribute to health-related community development, companies introduced the concepts of an ecosystem of giving and leveraging core business strengths, but not necessarily as expected. It was interesting that responses to questions about health-related initiatives mentioned that they would ideally coordinate with other companies and partners so as to compliment, rather than compete with one another's charitable giving initiatives. The concept of an ecosystem of giving among non-profit anchor institutions is not new [24]; we add to the literature that for-profit businesses have a similar view and do not want to overlap in giving initiatives with other companies. Further, representatives of non-health companies seemed to assume that health-related community development contributions would be best left to health industry companies because they would be better positioned to leverage core business strengths to maximize impact of contributions. In another study, for-profit hospitals' were indeed focused on impacting health-related initiatives in local communities to ultimately help their own business [25]. However, some businesses represented in our sample that were contributing to health-related community development leveraged strengths that were not health-related (e.g., market analysis). Therefore, there may be previously overlooked opportunities for companies and non-profits in a local community to collaborate and leverage potential anchor businesses' core business strengths to support health-related community development.

Another interesting finding was that health-industry companies in this small sample took different approaches to health-related community development. One of the two health care industry companies was motivated to contribute to health-related community development to leverage business strengths. On the other hand, companies in health care industries may perceive conflict of interest as a barrier to contributing to health-related community development, as demonstrated by another health care company in this interview sample. As our sample size is small, more work on health industry potential anchor businesses is needed to reach data saturation regarding these perspectives. Meanwhile, health care industry companies might examine ways to address companies' concerns around conflict of interest such as working with non-profits.

The interviews of these potential anchor businesses revealed that additional research is needed to examine if being "anchored" to a local community is associated with contributions to community development. A company's mobility and economic ties to the community have been proposed as part of the definition of an anchor business. However, it was not clear in this small interview sample, based on company representatives' descriptions, whether mobility or a

company's economic ties to the area were related to the company's willingness to contribute to community development. Further, few companies in our sample had policies to source from local suppliers, which could mean that these policies are not public, or unlike "eds and meds", supplies for these for-profit business operations are not available locally. Future work is needed to determine if the type of business operations influences if potential anchor businesses choose local suppliers and contracts.

All companies that we interviewed that we considered to have substantial investments in community development had direct-to-consumer business models and the low-contributors did not. Further, the low-contributors were dominated by the founder families which may influence the company philosophy. This may be a worthwhile subject for future study to further elucidate if the pathway of a company's goods or services to consumers as well as the businesses' mobility and economic ties predicts willingness to engage in community development initiatives as well as contribution generosity.

## 4.1. Limitations

Our screening criteria and thus our sample may not be representative. We selected companies based on the relative economic impact by revenue or employment, rather than the absolute revenues or profitability. Further, we used thresholds of ratios determined using example anchor businesses. That said, a strength of this study was the national random sampling of companies with economic ties to a local community similar to that of accepted anchor businesses. Our sample is more representative than prior case studies, or studies using convenience samples or targeting the largest companies or particular industries. Nonetheless, our sample was subject to non-response bias. Having recruited for interviews during the COVID pandemic, potential interviewees were working remotely and thus may have been more difficult to reach. Companies had closed, merged, and been purchased, and may have experienced staff turnover. Further, the positions of interest to this study may have been terminated, leading to higher non-response. Compared to respondents, non-respondents were larger companies in terms of revenue but contributed much less to community development. As a result, non-response bias limits the generalizability of our findings.

Further, there may be more efficient groupings of companies than as we did. In previous work, it has been determined that corporate social responsibility is more common in industries that rely on advertising and marketing to generate revenue [26]. We did not have a sufficient sample to examine whether our results differ, for example, between companies in specific industries, or in healthcare vs. non-health industries or by the amounts donated to their company foundations.

A limitation of our work was that we determined what was considered to be a threshold for substantial investment, which influenced our stratified sampling for the interviews, was defined empirically rather than relative to either company revenues or relative to the revenue of the local community. There are arguments for both of these measures may have improved our interpretation of who contributed to the respective community. Further, excluding company foundations may also be a limitation, though as we mentioned above, foundations can have financial support from beyond the company itself.

For this study, we did not track the focus of specific community development initiatives other than grouping them by health and non-health issues. This is a topic for future work as for example, there is increasing awareness and community development focused on racial equity and climate change. As for-profit entities, there may be tax incentives or capital structures in place that favor some types of community development over others. A larger future

study might address for-profits' motivations to contribute to specific health and non-health related topics.

## 5. Conclusions

In summary, potential anchor businesses are widely distributed across the U.S., with about one third of counties having at least one such business. Our findings about motives suggest that the generosity of contributions from potential anchor businesses to community development was largely determined by their philosophy about its relationship with the community. Companies driven primarily by commercial motives tend not to contribute substantially to community development. Contributions to health-related community development initiatives were often associated with core business strengths, whether or not those strengths were health-related.

Our findings may be utilized by foundations and policymakers who are interested in spurring contributions by potential anchor businesses to community development in general and, in particular, health-related initiatives. For example, the companies that satisfy our screening criteria as potential anchor businesses may be a good resource to identify such candidates. Specifically, an examination of companies' philosophies regarding community development contributions could help further focus on those that tend to contribute substantially. Effective communication and messaging can be developed to encourage potential anchor businesses that have not contributed substantially to leverage their business strengths to help develop surrounding communities. Further research is warranted to investigate how best to leverage potential anchor businesses' core strengths, avoid conflict of interest, and build partnerships with other organizations in the community for community development purposes.

## Supporting information

**S1 Fig. Geographic distribution of potential anchor businesses in the United States.**
(TIF)

**S1 File. Abstraction tool for the environmental scan.**
(PDF)

**S2 File. Interview guide: RWJF for-profit anchor business rationale.**
(PDF)

**S3 File. Environmental scan data.** We share data that are non-proprietary, deidentified and not identifiable by inference in combination with the qualitative interview data.
(XLSX)

## Acknowledgments

We would like to acknowledge our interview participants for taking the time to speak to us about their experiences. We would also like to acknowledge Christina Dozier and Megan Collado for their support in our recruitment efforts and RWJF grantees for their suggestions of exemplar anchor businesses and feedback regarding the potential anchor business operationalization definition.

## Author Contributions

**Conceptualization:** Catherine C. Cohen, Raymond Tsai, Harry H. Liu.

**Data curation:** Catherine C. Cohen, Harry H. Liu.

**Formal analysis:** Catherine C. Cohen, Nabeel Qureshi, Harry H. Liu.

**Funding acquisition:** Catherine C. Cohen, Harry H. Liu.

**Investigation:** Catherine C. Cohen, Nabeel Qureshi, Harry H. Liu.

**Methodology:** Catherine C. Cohen, Nabeel Qureshi, Harry H. Liu.

**Project administration:** Harry H. Liu.

**Writing – original draft:** Catherine C. Cohen, Nabeel Qureshi, Harry H. Liu.

**Writing – review & editing:** Catherine C. Cohen, Raymond Tsai.

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
