## [Decision Letter · Decision Letter 0]

8 Mar 2022

PONE-D-22-02864Motivations of anchor businesses to support community development and community healthPLOS ONE

Dear Dr. Cohen,

Thank you for submitting your manuscript to PLOS ONE. After careful consideration, we feel that it has merit but does not fully meet PLOS ONE’s publication criteria as it currently stands. Therefore, we invite you to submit a revised version of the manuscript that addresses the points raised during the review process.

We look forward to receiving your revised manuscript.

Kind regards,

Dragan Pamucar

Academic Editor

PLOS ONE

Journal Requirements:

Reviewers' comments:

Reviewer's Responses to Questions

**Comments to the Author**

1. Is the manuscript technically sound, and do the data support the conclusions?

Reviewer #1: Yes

Reviewer #2: Partly

Reviewer #3: Yes

2. Has the statistical analysis been performed appropriately and rigorously? 

Reviewer #1: Yes

Reviewer #2: N/A

Reviewer #3: Yes

3. Have the authors made all data underlying the findings in their manuscript fully available?

Reviewer #1: Yes

Reviewer #2: No

Reviewer #3: Yes

4. Is the manuscript presented in an intelligible fashion and written in standard English?

Reviewer #1: Yes

Reviewer #2: Yes

Reviewer #3: Yes

5. Review Comments to the Author

Reviewer #1: I recommend to accept with minor revisions - manuscript could use some more detail/context, please see my attached comments. In general, I find the research adds to the body of knowledge on anchor institutions in the US.

Reviewer #2: I am very interested in the topic of anchor institutions and appreciate the opportunity to review this well written manuscript.

Introduction. The authors provide a nice summary of the literature in terms of definitions of anchors and anchor businesses. More effort needs to focus on the motivation for this study. They correctly note that much is known about drivers of business contributions to community development by for profit businesses, but less is known about anchor businesses. Why would the motivations of business anchors be different than general for-profit businesses, which also include business anchors? And there have been many representatives from anchor businesses who have discussed their organizations’ motivations for community contributions at conferences on this topic (National Academy of Medicine, RWJF Forums, US Chamber of Commerce Forums).

Methods. The methods section needs more detail. For example, what information does DonorSearch contain regarding nonpolitical and non-foundation contributions? The $50K threshold strikes me as very low, particularly when companies often point to their foundations as being the main source of community contributions, and those contributions can be millions of dollars for large businesses. For example, a $50K annual contribution from a large distribution warehouse employing potentially hundreds or thousands of workers strikes me as unsubstantial and not a meaningful investment in the community.

I’m wondering why the team included areas with > 1 million residents as businesses are less likely to have a local or reginal influence in these areas. Particularly if they are only contributing $50K.

I suspect think there’s a big difference in motivation toward community contribution between companies that are one of several within a community and those that are the only large business in the community.

“To confirm economic ties to the community, we collected information regarding whether the

company hired local, sourced their supplies locally, supported local business development or

participated in their local chamber of commerce.” Where did the authors collect this information from? I cannot imagine that this is available on websites beyond a vague/broad description. And what were the thresholds used?

Some of the information in the Methods section is vague. What data specifically came from SEC filings? What were the 25th percentiles for employment and revenue? It would be helpful if the team shared the data collection tool and protocol in an appendix.

Section 2.3, I would caution the authors against using the term “representative sample,” regarding the interviews.

Can the authors clarify whether they only interviewed one person per company? In looking at the interview protocol, I’m surprised that a single person could give correct answers to all of these questions. For example, “What percent of your company’s assets are invested in fixed assets, such as real estate, or infrastructure, in this community?” and “Please describe your company’s beliefs about corporate social responsibility and a business’ role in community development, if any?”

Also, how would the authors handle a company respondent who said that they were not in the top 10 employers in terms of influence on the surrounding communities (and therefore not meeting the anchor business definition)? Was there any basis for the respondents assessments?

Eds and Meds are often viewed differently than other anchors. Anchor businesses

Results

Looking through the results, I’m realizing that there may be a discrepancy between the giving profile of a company, and that of a branch/plant in a particular community. This is an issue for a large company headquarted in on area, but with branches/plants/warehouses in other areas. Are the dollars described in the manuscript specific to a specific community, or is that the total dollar value across the US (or internationally)? Is this study looking at headquarters only?

Some respondents were from philanthropy, but in the Methods section, philanthropy dollars were not included in the analysis.

Only 8 of 57 respondents agreed to participate? This strikes me as particularly problematic.

I thought the study was focused on for-profit anchor institutions, but some of the results appear to include “Eds and Meds”.

Overall, I’m a bit underwhelmed by the findings and have concerns about the methods and large number of non-participants.

Reviewer #3: Overall, this is a strong paper on an important topic. Clarity in details throughout the paper would help strengthen it further. Specifically, I would encourage the authors to include more context about the data sets used. Additionally, more context regarding the rationale for specifically including and identifying health-related initiatives would be helpful. The authors may also find recent work by Choyke/Cronin/Franz helpful in discussing health-related anchors. Please see additional, more specific points of feedback below:

L79-80 - Worth noting this quality isn't unique to for-profits; nonprofits often benefit similarly.

L120-121 - Recommend clarifying why organizations are considered anchors (e.g., large employer, economic impact, etc.).

L143 - One example of a database that should be explained in more detail (purpose, etc.).

L164 - More information regarding the DonorSearch database would be helpful as well.

L238-239 - The importance of the environmental sustainability policies could use clarification.

L243-244 - Examples would be helpful in understanding the nature of health references in mission statements.

L420-L425 - This paragraph is confusing - more clarification is needed to understand why health would not be a strength of a health contributor.

6. PLOS authors have the option to publish the peer review history of their article (what does this mean?). If published, this will include your full peer review and any attached files.

Reviewer #1: No

Reviewer #2: No

Reviewer #3: No

---

## [Author Response · Author response to Decision Letter 0]

22 Apr 2022

Please see the uploaded response to reviewers document.

---

## [Decision Letter · Decision Letter 1]

20 May 2022

Motivations of potential anchor businesses to support community development and community health

PONE-D-22-02864R1

Dear Dr. Cohen,

We’re pleased to inform you that your manuscript has been judged scientifically suitable for publication and will be formally accepted for publication once it meets all outstanding technical requirements.

Kind regards,

Dragan Pamucar

Academic Editor

PLOS ONE

Additional Editor Comments (optional):

Reviewers' comments:

Reviewer's Responses to Questions

**Comments to the Author**

1. If the authors have adequately addressed your comments raised in a previous round of review and you feel that this manuscript is now acceptable for publication, you may indicate that here to bypass the “Comments to the Author” section, enter your conflict of interest statement in the “Confidential to Editor” section, and submit your "Accept" recommendation.

Reviewer #2: All comments have been addressed

Reviewer #3: All comments have been addressed

2. Is the manuscript technically sound, and do the data support the conclusions?

Reviewer #2: Partly

Reviewer #3: Yes

3. Has the statistical analysis been performed appropriately and rigorously? 

Reviewer #2: I Don't Know

Reviewer #3: Yes

4. Have the authors made all data underlying the findings in their manuscript fully available?

Reviewer #2: Yes

Reviewer #3: Yes

5. Is the manuscript presented in an intelligible fashion and written in standard English?

Reviewer #2: Yes

Reviewer #3: Yes

6. Review Comments to the Author

Reviewer #2: Thank you for the opportunity to review a revision of this interesting paper. I think the authors thoroughly responded to the comments from the three reviewers. The paper still contains a number of sizable limitations (e.g., 8 of 57 possible respondents agreed to be interviewed) but the authors acknowledge the limitations, making them more transparent to readers.

Reviewer #3: Thank you for taking the time to address the provided feedback. I have no additional concerns at this time.

7. PLOS authors have the option to publish the peer review history of their article (what does this mean?). If published, this will include your full peer review and any attached files.

Reviewer #2: No

Reviewer #3: No

---

## [Editor Report · Acceptance letter]

6 Jul 2022

PONE-D-22-02864R1 

Motivations of potential anchor businesses to support community development and community health 

Dear Dr. Cohen:

I'm pleased to inform you that your manuscript has been deemed suitable for publication in PLOS ONE. Congratulations! Your manuscript is now with our production department. 

Kind regards, 

on behalf of

Dr. Dragan Pamucar 

Academic Editor

PLOS ONE